# Impact of Chitosan on Water Stability and Wettability of Soils

**DOI:** 10.3390/ma14247724

**Published:** 2021-12-14

**Authors:** Agnieszka Adamczuk, Milena Kercheva, Mariana Hristova, Grzegorz Jozefaciuk

**Affiliations:** 1Department of Physical Chemistry of Porous Materials, Institute of Agrophysics of Polish Academy of Sciences, 20-290 Lublin, Poland; jozefaci@ipan.lublin.pl; 2Department of Physics, Erosion, Soil Biota, Institute of Soil Science, Agrotechnologies and Plant Protection “N. Poushkarov”, 1331 Sofia, Bulgaria; mkercheva@abv.bg; 3Department of Genesis, Diagnostics and Soil Classification, Institute of Soil Science, Agrotechnologies and Plant Protection “N. Poushkarov”, 1331 Sofia, Bulgaria; marihristova@hotmail.com

**Keywords:** aggregate, chitin derivatives, soil reaction, water repellency, destruction kinetics, food wastes

## Abstract

Chitosan has become increasingly applied in agriculture worldwide, thus entering the soil environment. We hypothesized that chitosan should affect the water stability of soil. Since this problem has not been studied to date, we examined, for the first time, the influence of chitosan on the water stability and wettability of soil aggregates. The aggregates were prepared from four soils with various properties amended with different amounts of two kinds of powdered chitosan, and subjected to 1 and/or 10 wetting–drying cycles. The water stability was measured by monitoring air bubbling after aggregate immersion in water, and the wettability was measured by a water drop penetration test. The biopolymer with a lower molecular mass, lower viscosity, and higher degree of deacetylation was more effective in increasing the water stability of the soil than the biopolymer with a higher molecular mass, higher viscosity, and lower deacetylation degree. After a single wetting-drying cycle, the water stability of the soil aggregates containing chitosan with a higher molecular mass was generally lower than that of the soil; after ten wetting–drying cycles, the water stability increased 1.5 to 20 times depending on the soil. The addition of low-molecular-mass chitosan after a single wetting-drying cycle caused the water stability to become one to two hundred times higher than that of the soil. A trial to find out which soil properties (pH, C and N content, bulk density, porosity, and particle size distribution) are responsible for the effectiveness of chitosan action was not successful, and this will be the objective of further studies.

## 1. Introduction

Due to the increasing food demand of the growing human population, the recycling and utilization of food industry wastes has become an urgent task worldwide. The main waste product of the food industry is chitin, an N-acetylglucosamine polysaccharide present in crustaceans (shrimps, lobsters, and crabs), mollusks (oysters and squids), insects, and fungi [1]. Currently, the main commercial source of chitin comprises waste streams from the marine food industry—mainly the exoskeletons of crustaceans. The annual world production of eight million tons of crustaceans for human consumption was estimated in 2016, 40% of which comprised waste exoskeletons with a chitin content of 15–40% [2], which equates to about 1600 tons of chitin produced yearly [3]. For comparison, one should mention that chitin comprises the second largest renewable carbon source after lignocelluloses coming from wood and agricultural wastes, such as straw of various kinds and sugar cane bagasse [4]. Starting from the mid-eighties, chitin-originated materials became broadly applied in wastewater treatment, pharmacy, medicine, biotechnology, the textile and paper industry, and many others [5]. Due to their high availability, biodegradability, phosphorus and nitrogen content, non-toxicity, bacteriostatic properties, and low cost, chitin-derived substances have become, more recently, increasingly applied in agriculture [6,7,8]. For the vast majority of the above applications, solid chitin is transformed to chitosan by decalcification (acid treatment) and removal of the acetyl residues (alkali treatment). The term “chitosan” is not uniquely related to a defined compound, but to a group of commercially available copolymers that are heterogeneous for the deacetylation degree, molecular mass, polymerization degree, surface charge, and acid dissociation constant [9]. These different characteristics, the degree of deacetylation and the molecular weight in particular, differentiate the physicochemical properties of the substance and, in consequence, the mode of its applications.

Up to date, the main interest of agriculture-related studies has been mostly concentrated on the effects of chitosan on plants and pests. It was proved that chitosan exerts significant effects on plant development and the survival of abiotic stresses [9,10,11,12]. A stimulating effect of chitosan on plant growth, yield, and macronutrient (nitrogen and phosphorus) uptake was observed by Boukhlifi et al. [13] for wheat and potatoes, Silva et al. [14] for melon, Chen et al. [15] for begonia, and Chookhongkha et al. [16] for chili fruits and seeds. Seeds coated with chitosan have a better germination capacity [17]. Chitosan is used to mitigate the following soil and plant pathogens: bacteria [18,19], viruses [20,21], fungi [22,23], or nematodes [24]. Chitosan is a promising coating material for slow-release fertilizers [25,26,27,28]. Studies related to the soil environment are mainly directed to the elaboration of new composite or copolymer systems containing chitosan for improvement of the soil water-holding capacity [29,30,31,32,33,34,35] and for soil stabilization [36]. Chitosan is also applied for the removal of various types of contaminants from soils [7,37,38,39,40].

As was shown above, chitosan may be introduced into a soil in various ways. Despite the fact that it is hypothesized to impact the physical and physicochemical properties of soil, studies on the above problem are very rare. Particularly, we could not find any papers reporting an effect of chitosan on the water stability and wettability of soil, which are crucial to understand the vast majority of soil agricultural, geotechnical and environmental functions, and properties important for tillage, erosion, compaction, aeration, slaking, water and solute transport, root penetration, road and building construction, and many others [41,42,43,44]. Therefore the objective of this study was to evaluate the influence of chitosan on the water stability of soil. To do this, we selected two different chitosans and four different soils. At first, the physicochemical properties of the chitosans and of the soils were characterized extensively, and soil–chitosan aggregates were prepared. The effects of biopolymer concentration and soil–biopolymer contact time on the water stability and wettability of the aggregates were investigated.

The water stability was measured by monitoring air bubbling after aggregate immersion in water, which is probably the only method that allows the kinetic parameters of the destruction process for rapidly destroyed large aggregates to be estimated.

The wettability was assessed by a water drop penetration test (WDPT), which reflected the rate of water infiltration into the aggregate.

## 2. Materials and Methods

### 2.1. Soils

Soil samples were taken from upper 5–15 cm layers of four soils localized in East Poland, air dried and screened by a 1 mm sieve. The characteristics of the soil samples are presented in Table 1. These data include the following:⚬pH measured in 1:2.5 soil:water suspension after 5 min continuous stirring;⚬Particle size distribution determined for organic matter-depleted soil (H_2_O_2_) by sieving and the pipette method after chemical dispersion of soil sample in sodium pyrophosphate;⚬Particle density, PD, measured by helium pycnometry using Quantum Crome Ultrapycnometer 1000 (Quantachrome, Boynton Beach, FL, USA);⚬Total organic carbon content determined by dichromate digestion [45,46] and nitrogen content determined by Kjeldahl method.

### 2.2. Chitosans

Two different kinds of chitosan were used. The first, abbreviated as CS1, was provided by Sigma Aldrich (St. Louis, MO, USA) and the second (CS2) was provided by Beijing Be-Better Technology Co., Ltd. (Beijing, China). The basic properties of both chitosans are presented in Table 2. The data presented include the following:⚬Total carbon and nitrogen content and particle density determined similarly to soil analysis;⚬Degree of deacetylation (DD) calculated from the carbon/nitrogen ratio (C/N) using the following equation from Xu et al. [47]:
DD = 1 − (C/N − 5.14)/1.72(1)

⚬Average molecular weight (M) determined from viscometric measurements performed for series of CS1 and CS2 solutions of decreasing concentrations in 0.02 Mol dm^−3^ acetic acid/0.02 Mol dm^−3^ NaCl, at 24 °C, using Hoppler rheo-viscometer. The intrinsic viscosity (η_int_) was determined as follows:
η_int_ = lim*_c_*_→0_ [(η(*c*) − η_s_)/(η_s_ × f)](2)
where η(*c*) is the viscosity of the chitosan solution at a given concentration *c*, η_s_ is the viscosity of the solvent and f is the *w*/*w* fraction of the chitosan in the solution.

The average molecular weight was calculated from Mark–Houwink equation [48], as follows:
η = KM^α′^(3)
where η is the intrinsic viscosity, and K and α’ are constants for a given solute–solvent system.

The following K and α’ values evaluated by Varum and Smidsrod [49] were used:
K = 8.43 × 10^−3^; α′ = 0.93(4)

⚬Chitosan chain stiffness parameter (x) introduced by Kasaai [50], calculated as follows:
x = DA/(pH × µ)(5)
where DA = 1−DD is the acetylation degree [51] of the chitosan and μ is the ionic strength of the chitosan solution of a given pH.

⚬Contact angle (θ) measured on the pressed chitosan pellets using a DSO 100 automatic drop shape analyzer (KRUSS, Hamburg, Germany).

The biopolymer with lower DD is characterized by lower content of nitrogen due to the lower number of NH_2_ groups. The molecular mass of CS2 (with higher DD) is lower than for CS1, which is in line with Kofuji et al. [52] who indicated that progress in deacetylation process decreases the molecular weight. They also noticed that solutions of chitosan with higher molecular weight tended to have higher viscosity as was observed in this study. Chitosan with lower DD (higher DA) has stiffer chain conformation. The measured density of both chitosans is close to a value of 1.5 g cm^−3^ reported by Gierszewska-Druzynska et al. [53]. The water contact angle of CS1 is higher than that of CS2. The contact angle decreases in time due to wetting of the chitosan surface and soaking of the droplet into the pellet body, as illustrated in Figure 1.

The wetting pathways seem to differ for both materials. The CS1 appears to swell much more intensively than CS2. The contact zone of water drop and CS1 “grows up” and finally an embankment of the swollen chitosan forms around the droplet. Such occurrences are hardly recognizable for CS2.

### 2.3. Preparation of Soil–Chitosan Aggregates

The soil samples were air-dried and screened by 1 mm sieve (mesh 18). To minimize eventual effects of chitosan granulometric composition on soil properties when studied further, both CS1 and CS2 were screened by the following set of sieves: 0.177 mm (mesh 80), 0.105 mm (mesh 140) and 0.053 mm (mesh 270), and the final materials were composed from equal weights of 0.177–0.105 mm and 0.105–0.053 mm fractions. Carefully homogenized water-saturated pastes were prepared from mixtures of the soils and the chitosans. Distilled water was used for paste preparation. The content of CS1 and/or CS2 in the mixtures was 0 (control), 0.5, 1, 2, 4, and 8%. Spherical aggregates with 20 mm diameter were formed from the pastes using ordinary silicon forms sold in fishing stores to prepare fish bait. The first set of aggregates was prepared just after the paste preparation and then air-dried (one cycle of wetting–drying), and the second set from the pastes subjected to 10 wetting–drying cycles (6 days per cycle). All aggregates were then dried until constant mass in laboratory conditions (relative humidity around 60% and temperature around 25 °C). The aggregates are abbreviated further using the abbreviation of a given soil (see Table 1) followed by the number of wetting–drying cycles (e.g., POD1 and POD10 denote aggregates formed from podzol preconditioned with one and ten wetting–drying cycles, respectively).

### 2.4. Studies of Soil–Chitosan Aggregates

Bulk density (BD) of the aggregates was estimated for laboratory-dried specimens. The aggregate weight minus the moisture content was divided by the volume of the aggregate. The aggregate volume was established based on Archimedes’ principle. The aggregate was totally immersed in the mercury liquid by forcing the aggregate down to a constant depth using an iron wire formed into a conical spiral and the increase in the system weight after immersion was measured. Knowing the mercury density we calculated the aggregate volume (the volume of the spiral was of course subtracted).

Water stability of the aggregates was estimated from air bubbling after immersion using a method described by Jozefaciuk et al. [54], which is briefly outlined below. The aggregate was thrown into a vessel submerged in water and hanging on a scale pan, and time-dependent increase in the weight of the aggregate, Δ*w,* due to the evolution of entrapped air from the interior of the destructed aggregate (decreasing in buoyancy), was registered. Next, one calculated the dependence of the extent of destruction (α) defined as follows:
α = Δw ÷ (w_final_ − w _initial_)(6)
where *w_final_* is the weight of the submerged aggregate after termination of the destruction and *w _initial_* is its initial weight registered just after immersion.

The dependence of α on time gives a characteristic sigmoidal curve reaching a plateau at the moment when the aggregate is totally destroyed. The above water destruction curve is interpreted in terms of a shrinking sphere model with the following equation:
1 − (1 − α)^1/3^ = 1/*t*_d_ × *t*(7)
where *t* is the time of the process and *t*_d_ is the time needed to terminate the destruction of the aggregate (destruction time).

The data plotted in coordinates of Equation (7) give a straight line reaching the value of 1 − (1 − α)^1/3^ = 1 when *t* = *t*_d_ (when α = 1). The value of the destruction time depends both on the characteristics of the aggregated material and the size of the aggregate. In the shrinking sphere model, the destruction time is proportional to the initial surface of the aggregate (*S*_0_). Therefore the ratio of *t*_d_/*S*_0_ [s m^−2^], which can be read as time necessary to destroy the unit surface of the aggregate, characterizing the aggregated material regardless of the size of the aggregate, is used as water stability parameter. For measurements of the destruction curves of the studied aggregates we used EXPLORER^®^ ANALYTICAL EX324M balance provided by OHAUS (Parsippany, NY, USA) with time resolution equal to 1 s. The final curves, averaged from at least 6 most similar destruction curves selected from 10 replicates for each aggregate, are considered further. Such selection was performed to minimize effects of structural artifacts influencing the destruction. The value of *S_0_* was estimated for each aggregate from its mass divided by bulk density (assumed to be the same for each aggregate).

It is worth mentioning here that we also attempted to test water stability of aggregates using wet sieving method (measuring the size distribution of aggregates and their mean weight diameter (MWD), before and after the action of water). Three to five millimeter fraction sieved out from the crushed aggregates was studied; however, in most cases the destruction was very fast and the final MWD reflected the granulometric composition of soil–chitosan mixtures. In the wet sieving method some external energy (mixing) is given to the aggregates, which markedly increases their destruction rate as compared to undisturbed process conditions.

Water repellency (hydrophobicity) was measured by a water drop penetration test [55], modified to achieve more precise results. Four microliters of distilled water was settled onto a surface of the aggregate (flattened with fine sandpaper and dust removed with a blower) and the time of the whole drop soaking was read from a video registering the process. The measurements were performed in four replicates using a DSO 100 automatic drop shape analyzer (KRUSS, Hamburg, Germany). It is worth mentioning that our first idea was to measure contact angles, CA, of the soil and soil–chitosan mixture; however, due to very fast water infiltration into the aggregates this was not possible. The WDPT selection was a matter of choice, since CA and WDPT frequently do not correlate.

## 3. Results and Discussion

Changes in the bulk density of the soil aggregates, amended with various doses of the studied chitosans, are illustrated in Figure 2.

The addition of both chitosans causes a marked decrease in the soil bulk density, which may be a direct consequence of the low particle density of the chitosans in respect to the solid phases of the four soils (Table 1 and Table 2). However, loosening of the soil structure, due to the addition of coarse chitosan particles, is also possible. The wetting–drying cycles consolidate the structure of the soils containing chitosans. The average effect is similar for both kinds of chitosan. The gelling/solubilization of chitosan is possibly responsible for the above effect. Consolidation of the structure was also observed by Hataf et al. [36] after the addition of chitosan dissolved in acetic acid to a sandy soil. They stated that chitosan increases interparticle interactions, concluding that this mechanism depends on the water content. Under wet conditions, the biopolymer enhances the bonds between soil particles, and during dry conditions, the chitosan gel converts to fibers with very low mechanical strength.

Exemplary water destruction curves of the soil aggregates, showing changes in the extent of destruction of the aggregates over time, as well as the above data plotted in coordinates of Equation (7), are shown in Figure 3.

Similar curves to those in Figure 4 were obtained for the other soils containing both CS1 and CS2, which indicates that the shrinking sphere model can be applied to identify the water destruction of the studied aggregates. Taking the *t*_d_ values calculated from the slopes of the linear fits of the destruction data of Equation (7), plotted in coordinates (see Figure 3), and dividing them by the initial surface of each aggregate, *S_0_*, the values of the time necessary to destroy the unit surface of the aggregate, *t*_d_/*S*_0_ [s m^−2^], were calculated.

The water stability of the aggregates containing CS1, preconditioned with a single cycle of wetting–drying, is generally lower than for the aggregates of the control soils. After ten wetting–drying cycles, these aggregates became more water resistant, with the exception of the aggregates of fluvisol and umbrisol, which contained the maximum dose of CS1. The latter aggregates are more water resistant than their counterparts when subjected to the single wetting–drying cycle, but they are still less water resistant than the control soils. In contrast to CS1, the water stability of the CS2 containing aggregates reaches high water resistance after the first wetting–drying cycle, and it increases only slightly after the next nine wetting–drying cycles. The impact of high doses of CS2 on the water stability of fluvisol and umbrisol is extremely high. The *t*_d_/*S*_0_ values reach up to three thousand seconds per square centimeter, which means that the time of destruction of the aggregate with a 20 mm diameter is over 10 h, while the control aggregates need a few minutes to be destroyed.

At least a few mechanisms can be responsible for the above effects. The addition of coarse chitosan particles may lead to loosening of the soil structure and breaking of the distance-dependent interparticle bonds; thus, the soil becomes more susceptible to water destruction. On the other hand, the improvement in water resistance may be connected with the solubilization/gelling of chitosan and the gluing action of its colloidal particles on soil grains. The first mechanism seems to dominate in CS1-containing aggregates, particularly after a single wetting–drying cycle, and for the maximum CS1 doses in neutral and alkaline soils. The effect of the second mechanism should increase over time, since solubilization and gelling are time dependent. The increase in water stability over time was observed for all the aggregates. The gluing of soil particles by the jellified chitosan should also affect the mechanical stability of the aggregates. If the gluing action overcomes the soil material dilution by the chitosan, the mechanical stability should increase, which may also be connected with water stability changes. We intend to study this problem in the near future.

The differences in the effects of both kinds of chitosan on aggregate stability may be connected with their molecular characteristics. CS1, with a greater molecular mass, may dissolve slower, so the time effect of CS1 on water stability is lower than that of CS2, which has a lower molecular mass. The rapid solubilization of CS2 may also be a reason why, just after the first wetting, the soils admixed with CS2 reach almost maximum water stability. The differences in the swelling properties of CS1 and CS2 (CS1 swells markedly faster than CS2; see Materials and Methods section) may also influence the lower stability of CS1-containing soils.

The water drop penetration time for the studied soils, amended with different amounts of the studied chitosans, is shown in Figure 5.

As a consequence of their high contact angles, both chitosans make the soils more water repellent. The time needed for droplet penetration for the biopolymer with a lower DD (CS1) was shorter than for the material with a higher DD (CS2). These results are in agreement with Mucha et al. [56], who reported that the water sorptivity of chitosan films decreased with increasing DD. In general, water repellency increases with an increasing number of wetting–drying cycles. Similar mechanisms to those governing water stability may be responsible for water repellency. The highest effect of chitosan on increasing water repellency was observed in podzol. We think that this soil has the lowest surface area (due to the smallest content of clay and organic matter), and the molecules of the dissolved chitosan may cover the surface to the greatest extent, forming hydrophobic patches. We thought that the water stability of the aggregates may be governed by their wettability; however, no correlation between the aggregate destruction time and WDPT was found.

We tried to establish which properties of the soils correlated with their reaction with chitosan. To do this, we used a parameter expressing the maximum impact of chitosan on the water stability of the soil, which was taken as a ratio of the maximum value of *t*_d_/*S*_0_ for the chitosan-containing aggregate to the same value for the control soil. For CS1, this parameter was 5.7 (for podzol), 23.9 (for arenosol), 2.2 (for fluvisol), and 1.3 (for umbrisol). The same values for CS2 were 97.3,17.8, 83.2, and 72.4, respectively. The first candidate to govern the water stability of the soil–chitosan aggregates was the pH of the soil, since it affects the chitosan solubility and, according to Kaasai [50] (see Equation (5)), an increase in pH reduces the stiffness of chitosan chains. However, no correlation was found between soil pH and the water stability of chitosan-containing soils. The next candidates, the amount of clay fraction and/or organic carbon content, also did not correlate well with the water stability of soil–chitosan aggregates. It seems that the properties of chitosan are more important in governing the water stability of soil aggregates than the properties of the soil; however, studying more soils may clarify the above problem.

In summary, the effect of chitosan on the water stability and wettability of soils increased over time. The water stability and wettability of soil depended on the properties of the added chitosan. Stronger and faster action was noted for chitosan with a lower molecular mass, lower viscosity, and higher deacetylation degree. The above material improved the stability of the soil aggregates by 100 to 200 times after just one cycle of wetting–drying, whereas chitosan with a higher molecular mass, higher viscosity, and lower deacetylation degree reduced the water stability of the soil aggregates after a single wetting–drying cycle, and caused it to increase 1.5- to 20-fold after 10 wetting–drying cycles. The water stability of the soil aggregates (time of aggregate destruction in water) did not correlate with the wettability of the soil (water drop penetration time). Wetting–drying cycles consolidated the structure (increased the soil bulk density) of the soils containing chitosans. The effect of chitosan on the water stability and wettability of soils depended on the physicochemical properties of the chitosan. No correlations were found between soil pH, organic matter, or clay content and the water stability of soils containing chitosan.

## Figures and Tables

**Figure 1 materials-14-07724-f001:**
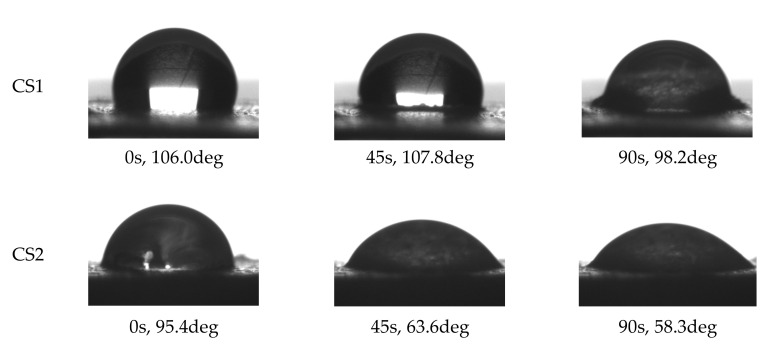
Time behavior of water drop settled on pellets prepared from CS1 and CS2. Below each photograph, time (seconds) and contact angle (degrees) are written.

**Figure 2 materials-14-07724-f002:**
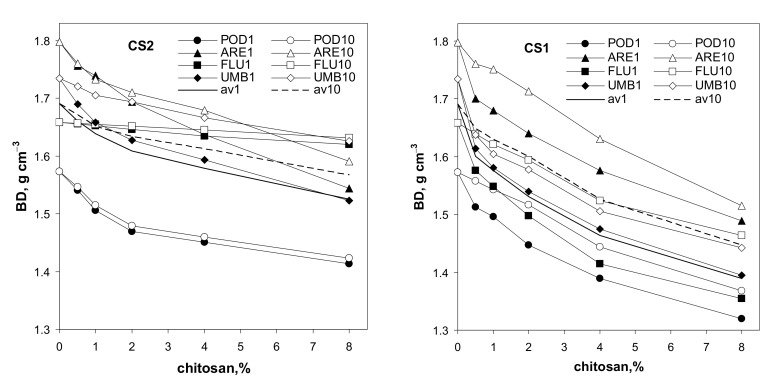
Dependence of bulk density of soil aggregates on amendment of the studied chitosans (CS1 and CS2). The soils are abbreviated as follows: POD—podzol, ARE—arenosol, FLU—fluvisol, UMB—umbrisol. The number after the soil abbreviation shows the number of wetting–drying cycles applied to soil aggregates. The curves denoted av1 and av10 show average data for all soils preconditioned with one and ten wetting–drying cycles, respectively. The error bars are covered by the labels of the points.

**Figure 3 materials-14-07724-f003:**
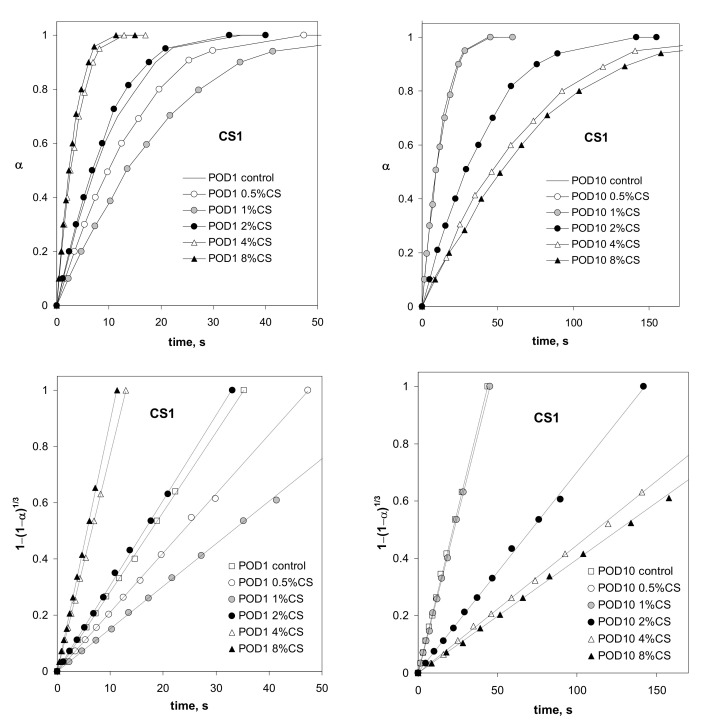
Time dependence of the extent of destruction of the aggregates of podzol (POD) amended with different amounts of chitosan CS1, preconditioned with one (left) or ten (right) cycles of wetting–drying (above) and the respective data, plotted in coordinates, of the shrinking sphere model according to Equation (7) (below).

**Figure 4 materials-14-07724-f004:**
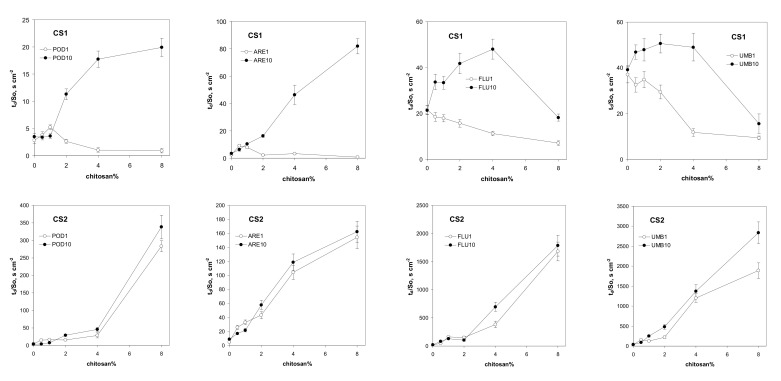
Dependence of time necessary to destroy the unit surface of the aggregate on the percentage of the added chitosan CS1 (above) and CS2 (below). The soils are abbreviated as follows: POD—podzol, ARE—arenosol, FLU—fluvisol, UMB—umbrisol. The number after the soil abbreviation shows the number of wetting–drying cycles applied to soil aggregates.

**Figure 5 materials-14-07724-f005:**
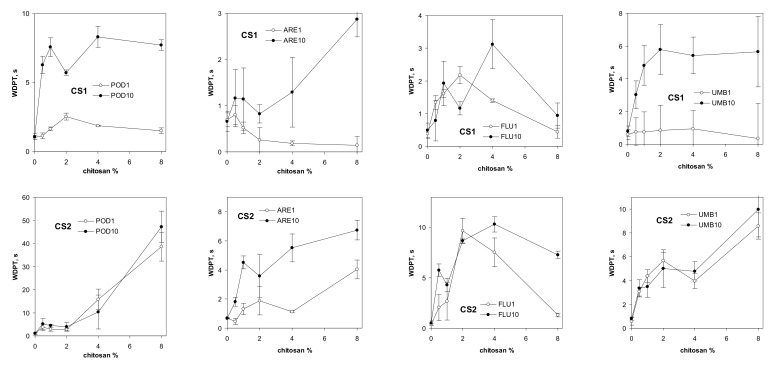
Dependence of water drop penetration time for aggregates of different soils on percentage of chitosan CS1 (above) and chitosan CS2 (below). The soils are abbreviated as follows: POD—podzol, ARE—arenosol, FLU—fluvisol, UMB—umbrisol. The number after the soil abbreviation shows the number of wetting–drying cycles applied to soil aggregates.

**Table 1 materials-14-07724-t001:** Characteristics of the studied soils.

Abbreviation	POD	ARE	FLU	UMB
Soil type	Podzol	Arenosol	Fluvisol	Umbrisol
Locality	Trawniki	Strzyzewice	Dorohucza	Prusy
Longitude E	22°58′41″	22°26′6″	22°59′38″	21°41′59″
Latitude N	51°9′14″	51°2′9″	51°9′43″	50°49′25″
pH	4.1	5.5	6.5	7.7
PD, (g cm^−3^)	2.54	2.62	2.62	2.68
Nitrogen (%)	0.16	0.13	0.46	0.14
Total organic carbon (%)	0.65	1.55	3.04	0.9
Sand (0.063–2 mm) (%)	72.4	47.1	20.2	10.4
Silt (0.002–0.063 mm) (%)	25.9	46.2	52.2	72.4
Clay (<0.002 mm) (%)	1.7	6.7	27.6	17.2

**Table 2 materials-14-07724-t002:** Properties of the studied chitosans.

	N [%]	TOC (%)	PD (g cm^−3^)	DD	M (kDa)	η (1% Solution) (cP)	x (at pH = 4, µ = 0.01)	θ (deg)
CS1	7.51	41.59	1.51	0.77	699	111.0	5.75	106.0
CS2	7.79	41.27	1.54	0.91	280	12.3	2.25	95.4

## Data Availability

All data are available from authors after a reasonable request.

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
