# Peer review of "Impact of Chitosan on Water Stability and Wettability of Soils"

_materials, 2021, doi:10.3390/ma14247724_

Round 1

Reviewer 1 Report

This paper presents a small study on the effect of chitosan on the stability and wettability of aggregates. The methodology followed here is well established, and there are related papers on chitosan. However, its effect on the stability and wettability of aggregates appears novel.

The writing is acceptable, even if some sentences may need readjusting. What is the meaning of “by its compulsive immersion in mercury” in L154?

My comments are on three major aspects:

  1. Aggregate stability and its measurement are well-established in soil science. Since the results are not conclusive on why chitosan reduces aggregate stability, it would be preferable to corroborate the results, by comparing them to another method(s).
  2. The authors measured wettability/water repellency by means of the WDPT. However, it only really measures the persistency of water repellency, which in this case is very small according to Fig 5. It would be preferable to measure the contact angles for the chitosan-soil mixtures (as in Fig 1). If for a flat surface you get 106deg, in chitosan-soil mixtures you could get even higher values (depending on whether it goes into a Wenzel or Cassie-Baxter state). Just follow Bachmann’s papers. CA’s and WDPT do not necessarily always correlate. Also, CA’s are a physical (usable) parameter, that could be related to all other soil properties (eg, hydraulic).
  3. The discussion on why the aggregate stability reduces with increasing chitosan content is interesting but inconclusive. My hypothesis is that the aggregate stability reduces because the aggregates are mechanically weaker. In soil mechanics, a reduction in bulk density (Fig 2) reduces material strength. If chitosan doesn’t undergo any form of polymerization with wetting-drying cycles, then increasing its content will reduce their aggregate strength.

Based on the above contents, the authors may consider generating new data.

Author Response

Reply on  Ref#1 comments

You wrote:

This paper presents a small study on the effect of chitosan on the stability and wettability of aggregates. The methodology followed here is well established, and there are related papers on chitosan. However, its effect on the stability and wettability of aggregates appears novel.

Thank You for this opinion.

The writing is acceptable, even if some sentences may need readjusting. What is the meaning of “by its compulsive immersion in mercury” in L154?

The aggregate of low density flows upon the mercury surface. To measure its volume, the aggregate should be totally immersed in the mercury liquid. To do this we forced the aggregate down to a constant depth using an iron wire formed into a conical spiral and measure the increase in the system weight after immersion. Knowing the mercury density we calculated the aggregate volume (the volume of the spiral was of course subtracted).  It is explained in the text.

My comments are on three major aspects:

  1. Aggregate stability and its measurement are well-established in soil science. Since the results are not conclusive on why chitosan reduces aggregate stability, it would be preferable to corroborate the results, by comparing them to another method(s).

Many methods established long time ago are still being used for testing water stability of aggregates as e.g. wet sieving (measuring the size distribution of aggregates - and their mean weight diameter - before and after the action of water), slaking loss (slaking index estimated from visual observations of aggregate breakdown in water after defined time periods), destruction under falling water droplets (assessment of the number of water droplets of defined energy needed to complete aggregate breakdown) or destruction in water (measuring time required to complete the dispersion of a soil aggregate immersed in water). All these methods describe rather a result of the aggregates destruction, mostly at arbitrarily defined conditions and not the process itself. They are considered to be not more than semi-quantitative and none of them has been commonly accepted. All methods of determination of water stability of aggregates give significant differences between the results of each other. Comparison of materials described by stability indices derived from different methods is limited or impossible. Newer, more practical and precise methods for determining soil aggregate stability were also proposed. Laser diffractometry measurements of aggregates suspended in water recording the dynamics of changes in the median of the particle-size distribution is applied. In the above method only aggregates smaller than 2 mm (3.5 mm in the newest equipment) can be studied. In this method, similarly to wet sieving, some external energy (mixing) is given to the aggregates that markedly increases their destruction rate as compared to the natural conditions. The mass loss of an aggregate immersed in water has also been used. In the above method the recorded change in the aggregate mass during the destruction resulted from two phenomena: unsticking of particles (decrease in the aggregate mass) and air evolution (decrease the buoyancy and the mass increase) and the simultaneous effect of both processes may lead to derive not real reaction destruction rates. Wet aggregate stability of soils from percolated soil aggregate columns measurements by 1H-NMR relaxometry was applied also, however only the percentage of water-stable aggregates could be determined accurately. Therefore we used a simple method that allowed for quantitative analysis of aggregate destruction in water basing on measurements of air bubbling process after aggregate immersion. We wanted to compare our results with wet sieving, however the aggregates disruption was too fast in most cases and we did not rely on the results.

  1. The authors measured wettability/water repellency by means of the WDPT. However, it only really measures the persistency of water repellency, which in this case is very small according to Fig 5. It would be preferable to measure the contact angles for the chitosan-soil mixtures (as in Fig 1). If for a flat surface you get 106deg, in chitosan-soil mixtures you could get even higher values (depending on whether it goes into a Wenzel or Cassie-Baxter state). Just follow Bachmann’s papers. CA’s and WDPT do not necessarily always correlate. Also, CA’s are a physical (usable) parameter, that could be related to all other soil properties (eg, hydraulic).

It was our first idea to measure CA’s of the soil and soil-chitosan mixtures. We used the same KRUSS apparatus as for CA measurements for chitosans and registered films illustrating the behavior of water drop in time. However due to very fast water infiltration into the aggregates this was not possible. Therefore from the above films we established WDPT. Of course CA and WDPT commonly do not correlate. Possibly we could apply a thin column or thin layer wicking procedure introduced by Van Oss to evaluate water contact angles and surface free energies of the studied mixtures. Since this method is extremely time and work consuming (according to our experience, few months would be necessary to complete the measurements) we are not able to do this in limited time given by the Editor to correct the manuscript.

  1. The discussion on why the aggregate stability reduces with increasing chitosan content is interesting but inconclusive. My hypothesis is that the aggregate stability reduces because the aggregates are mechanically weaker. In soil mechanics, a reduction in bulk density (Fig 2) reduces material strength. If chitosan doesn’t undergo any form of polymerization with wetting-drying cycles, then increasing its content will reduce their aggregate strength.

Again our thoughts go by similar pathways. We almost completed the measurements of the mechanical strength of the same soil-chitosan aggregates to check if chitosan undergoes any form of polymerization with wetting-drying cycles. We want to publish these data in a near future and correlate them with water stability.  We mentioned a possible connection of water and mechanical stability of aggregates in the text by adding a sentence:Gluing of soil particles by the jellified chitosan should also affect a mechanical stability of aggregates. If gluing action overcomes soil material dilution by the chitosan, the mechanical stability should increase, that may also be connected with water stability changes. We like to study this problem in near future.”

Based on the above contents, the authors may consider generating new data.

Unfortunately, generating new data as suggested in points 1. was not successful. Fulfilling point 2. is not possible in a reasonable time. To complete point 3. we need 2-4 months. Therefore, if You don’t mind, we like to leave the manuscript in the present form (all authors want to publish new findings as early as possible).

Yours sincerely, the authors.

Reviewer 2 Report

A very interesting and important study that contributes to the expansion of knowledge about the safety of chemicals for the environment. Overall, the manuscript is interesting and well presented, but there are a few things that could be improved.

1. In the abstract, it is recommended to add numbers (or a range of values) indicating how much chitosan application changes soil water stability.
The research plan is not described. It is recommended to add it before the "Materials and Methods" section.

2. "2.1. Preparation of soil / chitosan aggregates" please explain why the chosen content of CS1 and CS2?

3. "3. Results and discussion" Are there any errors in the data presented in Figures 2 - 3? If there are, maybe they should be indicated?

4. Summarizing: The conclusion about how chitosan affects soil water stability is not very clear. It might be worth adding specific values ​​in%.

Author Response

Reply for Ref#2 comments

You wrote:

A very interesting and important study that contributes to the expansion of knowledge about the safety of chemicals for the environment. Overall, the manuscript is interesting and well presented, but there are a few things that could be improved.

Thank You for this opinion.

  1. In the abstract, it is recommended to add numbers (or a range of values) indicating how much chitosan application changes soil water stability.

We added a fragment: After a single wetting-drying cycle, water stability of soil aggregates containing chitosan of higher molecular mass was generally lower than that of the soil, whereas after ten wet-ting-drying cycles water stability increased from 1.5 to 20 times depending on the soil. One to two hundred times higher water stability than that of the soil was caused by addition of low molecular mass chitosan just after the single wetting-drying cycle.

The research plan is not described. It is recommended to add it before the "Materials and Methods" section.

We rewrote the last phrases of the Introduction to fulfill this request: To do this we selected two different chitosans and four different soils. At first physico-chemical properties of the chitosans and of the soils were characterized extensively and soil-chitosan aggregates were prepared. Effects of biopolymer concentration and soil-biopolymer contact time on water stability and wettability of the aggregates was investigated.

  1. "2.1. Preparation of soil / chitosan aggregates" please explain why the chosen content of CS1 and CS2?

In preliminary experiments we observed that chitosan addition up to 10%w/w exerts the most visible effects on water stability.

  1. "3. Results and discussion" Are there any errors in the data presented in Figures 2 - 3? If there are, maybe they should be indicated?

The precision of the BD measurements was very high. In Figure 2 the error bars are covered by the labels of the points. It is mentioned in the Figure caption. Since Figure 3 contains exemplary results, error bars are not necessary.

  1. Summarizing: The conclusion about how chitosan affects soil water stability is not very clear. It might be worth adding specific values ​​in%.

We enlarged the description of chitosan effect on water stability, similarly as it was done in the abstract.

Yours sincerely, the authors.

Reviewer 3 Report

The abreviated name of the chitosans are C1 and C2 or CS1 and CS2? The solubility should be gived for the chitosans. The dry density of the soil/chitosan aggregates have influence to the water stability so should be controled. 

Author Response

Reply for Ref#3comments

The abreviated name of the chitosans are C1 and C2 or CS1 and CS2?

CS1 and CS2, It is corrected in the whole text.

The solubility should be given for the chitosans.

Chitosan is not soluble in water. In some other solvents it jellifies and so “solubility” has no physical sense. Therefore instead of solubility, the viscosity of the chitosan gel of defined concentration is used, as it was done in the manuscript.

The dry density of the soil/chitosan aggregates have influence to the water stability so should be controled. 

This was what we did. The bulk density measured by us is a dry density of the aggregates.

 Yours sincerely, the authors.

Round 2

Reviewer 1 Report

The authors haven't addressed my comments. The authors have only justified their experimental rationale, with no new information added to the manuscript. My questions on the aggregate stability and wettability remain. Suggest the paper to be returned to the authors.

Author Response

Reply on  Ref#1 second comments

You wrote:

The authors haven't addressed my comments. The authors have only justified their experimental rationale, with no new information added to the manuscript. My questions on the aggregate stability and wettability remain. Suggest the paper to be returned to the authors.

We addressed Your comments in the manuscript. The respective fragments are written in blue.

Yours sincerely, the authors.